# The Roman Catholic Parish in the Face of the Ukrainian Refugee Crisis: A Case Study of St. Joseph Parish in Chorzów, Poland and Holy Spirit Parish in Kátlovce, Slovakia

Rafał Śpiewak [1,*], Wiktor Widera [2], Denisa Jánošová [3] and Tomasz Jobczyk [3]

[1] Department of Market and Marketing Research, University of Economics in Katowice, Katowice, 1 Maja 50, 40-287 Katowice, Poland

[2] Faculty of Social Science, University of Silesia in Katowice, Bankowa 12, 40-007 Katowice, Poland; wiktor.widera@us.edu.pl

[3] Faculty of Mass Media Communication Námestie Jozefa Herdu, University of Ss. Cyril and Methodius in Trnava, 577/2, 917 01 Trnava, Slovakia; denisa.janosova@ucm.sk (D.J.); jobczyk1@ucm.sk (T.J.)

[*] Correspondence: r_spiewak@onet.eu

**Abstract:** Russia's war against Ukraine as a result of the Kremlin's aggressive policy has caused a number of catastrophic material consequences, but most dramatically, the suffering of the Ukrainian people. Millions of people at risk of death were forced to leave their previous places of residence. In a short space of time, they had to make the crucial decision to leave the war-stricken country without the prospect of an exact return date. They were accompanied not only by the uncertainty of finding a safe and friendly place to survive but also by the fear of whether they would have anything to return to in the future. From the first days of the war, huge migrations began inside and outside Ukraine. This dramatic situation of an unprecedented wave of refugees in 21st-century Europe has mobilized societies and governments in many countries to provide unprecedented assistance to the citizens of Ukraine. The largest number of refugees reached Poland. In addition, a significant number went to Slovakia. This article, the result of a collaboration between Polish and Slovakian researchers, attempts to illustrate the response of Catholic Church parishioners to the influx of refugees from Ukraine using the example of the St. Joseph parish in Chorzów and the Holy Spirit parish in Kátlovce, Slovakia. The parishioners communities undertook charitable activities, and an informal aid group was immediately organized in the Polish parish. The aim of the research process carried out was, on the one hand, to identify the forms of aid implemented, its scope, and the difficulties encountered, and, on the other hand, to try to grasp the motives for involvement in helping refugees. An important aspect of the research was to verify the reactions of the parishioners in the context of the guidelines of the social teaching of the Church, especially the teaching of Pope Francis on helping migrants and refugees.

**Keywords:** pastoral care; parishioners; military conflict; refugee; immigration; social teaching of the church; St. Joseph parish in Chorzów; Holy Spirit parish in Kátlovce





## 1. Grounds for Addressing the Issue

Between 24 February and 1 November 2022, 7.429 million refugees from Ukraine crossed the Polish-Ukrainian border, according to a communiqué by United Nations High Commissioner for Refugees (UNHCR) Filippo Grandi[1]. As a result of the war in Ukraine, the number of refugees worldwide has exceeded 100 million for the first time in history[2]. A total of more than 5.630 million people have returned to Ukraine since 24 February[3]. The largest group among those arriving in Poland were women, most often mothers with children[4]. The Polish state immediately took a number of administrative measures to enable the necessary assistance. The legal basis for the Polish state's assistance to take organizational measures was the adopted special law on assistance to refugees from

Ukraine. According to its provisions, Ukrainian citizens were allowed to stay legally on the territory of the Republic of Poland for a period of 18 months, with a simultaneous indication that this period could be extended for another 18 months. Assistance activities were organized on the territory of local government units, to which the Act enabled the Polish government to declare budgetary support necessary for the implementation of activities in the area of, inter alia, access to education for citizens of Ukraine and especially access to health care.

However, the situation showed that the greatest burden of helping the refugees fell on Polish society, which organized the needed assistance in various ways. This process revealed the enormous activity and commitment of Polish citizens. The scope of the tasks sometimes involved serious, unplanned expenses in household budgets. This was particularly the case when Ukrainians had to be taken into their own flats and houses. This challenge was declaratively secured by the state in the form of financial support in the amount of 40 PLN per day per person accommodated in a household. This assistance was provided for a period of two months. As part of its assistance activities, the Government of the Republic of Poland enabled Ukrainian children and young people to continue their education in Polish schools. Special reception points were opened for children from orphanages. A key action of a formal nature was the creation of the possibility for Ukrainian citizens to obtain PESEL numbers in order to simplify their handling of everyday matters necessary during their stay in Poland. Refugees whose stay in Poland was recognized as legal, confirmed by the granting of a PESEL number, were entitled to state aid in the form of a single cash benefit of PLN 300 per person[5]. In addition, they also had legal access to the labor market, education, and health services.

A similar situation occurred in Slovakia. This country, which borders Ukraine, was also involved in helping war refugees. Those hosting Ukrainians received support from the state. Within a short time of crossing the border into Slovakia, Ukrainian refugees were given accommodation, the right to work, health insurance, and the right to social benefits. Refugees were also granted asylum status. The main difference between temporary shelter and asylum was that those applying for it under special conditions did not have to wait months to be granted it, as in the case of the standard asylum procedure. Individuals and institutions that decided to host refugees from Ukraine could count on material support from the state. This included individuals, owners of guesthouses and hotels, as well as local authorities, charitable organizations, and church institutions[6]. All information for Ukrainians arriving in Slovakia was posted, among other things, on a specially launched website[7]. In addition, a portal with job offers was also launched on the website of the Centre for Labor, Social Affairs, and Family of the Slovak Republic[8].

The main identified problem for refugees from Ukraine was their lack of ability to adapt to conditions in a foreign country, compounded by their lack of language skills. Accompanied by trauma and various health and mental health problems upon arrival in Poland and Slovakia, they were accompanied by fear, sometimes distrust of people, feelings they did not know so far, and a feeling of helplessness. A very serious worry was the uncertainty about the fate and even the lives of relatives left behind in Ukraine, especially the elderly or men mobilized for military service[9]. Polish and Slovak society at this particular time showed above-average sensitivity to the fate of refugees affected by the drama of war. The scale of this phenomenon has been recognized by various political circles or institutions in Europe. The dissemination of examples of aid through Polish and foreign media messages, however, was rarely supported by invoking the motives of solidarity initiatives. This aspect is of particular importance for the formation of the content of educational processes, which in turn translates into the process of shaping universal values that influence the moral condition of society. From this perspective, the behavior of communities receiving refugees can be analyzed in terms of worldview determinants that influenced the attitude they adopted towards humanitarian challenges caused by the sudden influx of refugees from Ukraine. From a historical and sociological point of view, taking into account the Polish and Slovakian circumstances, the most influential doctrine

seems to be the disseminated social teaching of the Catholic Church. Following the thought of Nancy Foner and Richard Alba, "religious organisations provide an all-encompassing system of belief, as well as a community where immigrants gather and form networks of mutual support with co-ethnics, they provide a psychological ballast helping to ameliorate the traumas of early settlement and frequent encounter with discrimination" (Foner and Alba 2008). Hence, this area was the focus of the researchers' attention.

## 2. Assistance from the Catholic Church

In the first period of the mass arrival of refugees from Ukraine in Poland, many institutions and non-governmental organizations, in addition to the state, took action in the area of necessary assistance in various ways. The Catholic Church also came to the rescue, both on an institutional and social level.[10] The aid was primarily carried out by the structures of Caritas Polska,[11] the largest Polish charity organization, which reports directly to the Polish Bishops' Conference.[12] This organization was called in this context the "rapid reaction force of the Church".[13] Already in the first days, it sent more than 150 lorries with humanitarian aid to Ukraine. It concerned mainly food, clothing, hygiene products, and medicines. The organization prepared 2000 places for children with disabilities and orphans. In a very short time, about 650 children found care in the homes of Polish families. Reception points were organized (ibid.). Caritas was also involved in direct aid activities on the Polish-Ukrainian border, setting up the so-called Tents of Hope equipped with stalls with hot meals and providing the necessary assistance right after crossing the border. Diocesan Caritas, subordinate to individual bishops throughout Poland, has become a key organizer of aid in local communities. Particularly in dioceses directly bordering Ukraine.

It became extremely valuable for parish communities to undertake immediate aid actions, which already at the first stage of the hostilities in Ukraine organized collections of the most needed items as well as collections of money that went to the account of Caritas Poland. A key aspect of the assistance, however, was the organization of the refugees' stay within the individual parishes. Both in the infrastructure of the parishes themselves and in the homes of parishioners. To a large extent, the cost of living for refugees was covered by specially organized cash collections carried out by individual parishes. To illustrate regional aid, the example of the Archdiocese of Cracow can be cited, where several thousand permanent or temporary accommodation places have been organized, in parish buildings, religious houses, directly in churches and vicarages, as well as in retreat houses.[14] This shows how the Polish state was able to accommodate such a large number of refugees, especially in such a short time.[15]

Fr Jozef Haľko, President of the KBS Council for Migrants and Refugees, addressed the faithful on behalf of the Catholic Church in Slovakia. He declared that the Catholic Church in Slovakia wants to be ready to help Ukrainians, as he pointed out, for evangelical reasons. He also appealed to the parishioners to help refugees.[16] As many Slovaks expressed their willingness to provide accommodation for the refugees, the Slovak Catholic Charitable Organization organized the registration of accommodation and a special financial collection to help the refugees. Many other non-profit organizations also joined the campaign, as did the Evangelical Church of the Augsburg Confession in Slovakia.[17] The National Blood Transfusion Service of the Slovak Republic also organized blood donations for victims of the war in Ukraine.[18]

Slovak Catholic Charities ran various collections and organized other forms of assistance for Ukraine.[19] For Ukrainian refugees in need, Slovak Catholic Charity has set up 25 support centers. The Slovak Episcopal Conference offered spiritual assistance to Ukrainians. The Spiš Catholic Charity organized assistance on both the Slovak and Ukrainian territories.[20]

Among the socially active Polish and Slovak parishes are the parish of St. Joseph in Chorzów in the Katowice archdiocese[21] and the parish of the Holy Spirit in Kátlovce in the Trnava archdiocese in Slovakia.[22] The activities of their parishioners are described in this

article. The hypothesis in the research process carried out is that the organized support activities of the faithful exemplify the reception of the social teaching of the Catholic Church, which has become the overriding motivation for undertaking humanitarian action.

## 3. Church Pastoral Care Open to Social Challenges

The central documents of the Second Vatican Council, the Dogmatic Constitution on the Church and the Pastoral Constitution on the Church in the Modern World, were clearly pastoral in nature (Bourgeios 2001, p. 7). The pastoral dimension of the Church, therefore, makes it necessary to read the signs of the times in the light of Revelation. The encounter with the living God that is guaranteed by the mission of the Catholic Church always takes place in a historical and concrete reality in which we can read the economy of salvation (ibid., p. 9). Following the thought of Hans Urs von Balthasar, who believes that the culmination of all Christian revelation is a society, God cannot be discovered except in the social dimension, that is, in the mystery of the community of persons (von Balthasar 1993). Thus, it can be said that the entire Christian mystery has an interpersonal and, at the same time, public dimension. Consequently, experiencing and contemplating God demands community and, consequently, sensitivity to the other person and his plight. Pastoral theology aims to grasp the connection between the revelation of God's mystery and His plan and the practical conditions of life at a particular moment of realization of this plan. This requires constant spiritual and intellectual sensitivity to read God's guidance in the concrete situations in which the Church community exists. Deep and conscious living of the gospel should also materialize in the public sphere when evangelical principles permeate social life. This, in turn, requires the constant application of the principle developed on the basis of Catholic social teaching: see—judge—act[23]. This is especially true of the basic community in the structure of the Church, which is the parish. As the CCC states, "The parish introduces the Christian people to participate in liturgical life and gathers them in celebration; it proclaims the saving doctrine of Christ; it practices the love of the Lord in good and fraternal works" (2179). From the above, it is clear that participation in liturgical life and exploring the Gospel of Jesus has a practical orientation and is verified in fraternal love. One of the most important pastoral goals is the application of the unchanging principles of the gospel in the context of a particular vision of society. Pastoral openness and vigilance are crucial at this point. It should be constantly inspired by reflection on the problems of today's world.

This is the conviction expressed by Pope Francis when speaking about the mission of today's parishes: "Sometimes I think we should put a sign on the door of the parish that says 'free admission.' Parishes should be close communities, without bureaucracy, focused on people and a place to receive the gift of the sacraments. They must once again become schools of service and generosity, whose doors are always open to the excluded. And for those who have been included. For all"[24]. He believes that "parishes are not clubs for the few that give a certain social affiliation" (ibid.). In this regard, he asks for courageous reflection on developing a new style of functioning in parish communities. Pope Francis expresses his conviction that parishes should renew the concept of communion, both with God and between people; he wants them to be more and more communities of faith, fraternity, and welcoming the most needy (ibid.).

The Church's community, finding inspiration and moral imperative in Revelation, seeks diligently to read the challenges posed by reality. The Church has often had to intervene in urgent matters of social life.[25] This has been done essentially on the basis of the guidelines that flow from the social teaching of the Church, which, according to the 1986 instruction of the Congregation for the Doctrine of the Faith, *Libertatis constientia* [no. 72] (On Christian Liberty and Liberation): "was born out of the encounter between the Gospel message and its requirements, expressed succinctly in the supreme commandment of love of God and others and justice, and the problems arising from the life of society" (Pyszka 2003, p. 13). Pope Francis' teaching on the issue of migrants and refugees has been a special topic provoked by the international situation since the beginning of his pontificate. Pope

Francis devoted much attention to this phenomenon during his pilgrimage to Poland in 2016. Although his teaching had a different historical context at the time, it seems that it can be taken and read as a kind of preparation for the unexpected events that followed a few years later. The outbreak of war in Ukraine caused by the Russian Federation's aggression against the country triggered an unprecedented exodus of civilians seeking refuge and assistance in neighboring countries, especially Poland (number of refugees). He expressed his concern for refugees in his teaching contained in the encyclicals "Fratelli tutti" and "Laudato Si". In the encyclical "Fratelli tutti", where he drew attention to the existing wars in various parts of the world. He considered every war to be a failure of politics and humanity (FT 261). In his view, they cause immense suffering to innocent civilians and, in a social and political sense, leave the world in a worse situation than the original one. The drama of war is setting in motion huge migration processes today. In a very concrete way, Pope Francis writes: "Let us turn our attention to the refugees, to those affected by nuclear radiation or chemical attacks, to the women who have lost their children, to the children maimed or deprived of their childhood. Let us pay attention to the truth about these victims of violence, let us look at reality through their eyes and listen to their stories with an open heart. In this way, we will be able to see the abyss of evil at the heart of war, and we will not mind being treated as naïve because we have chosen peace" (FT 261).

According to Pope Francis, a constitutive feature of civil society is a sense of responsibility for fellow human beings (LS 25). He expressed this view in the context of climate change, which is becoming an acute global problem and is causing a tragic increase in the number of migrants. The Pope states: "They carry the burden of their lives, left to their own devices, without any norms of protection. Unfortunately, there is widespread indifference to these tragedies that are happening in various parts of the world" (LS 25). Awareness of the existing phenomena of migration and refugeeism should give rise to appropriate initiatives at various levels of society. In view of the enormity of the suffering, the urgency to help should arise. As he states, "This entails taking certain necessary measures, especially in the case of those fleeing major humanitarian crises. For example: increasing the number of visas granted and simplifying procedures; applying private and community sponsorship, opening humanitarian corridors for the most vulnerable refugees; providing adequate and decent accommodation; ensuring personal safety and access to basic services; ensuring adequate consular protection, the right to carry personal identity documents with them at all times, equal access to justice, the ability to open bank accounts and the provision of the necessities of life; ensuring freedom of movement and the possibility to work; protecting minors and providing them with regular access to education; providing temporary care or reception programmes; ensuring religious freedom; promoting their social integration; fostering family reunification and preparing local communities for integration processes" (FT 130). His teaching goes hand in hand with taking appropriate action. Pope Francis, with the papal letter *Humanam progressionem*, signed with the date 17 August 2016, created the Dicastery for the Promotion of Integral Human Development with a special section for Migrants and Refugees. The statutes of the new dicastery were approved on 17 August 2016 by the Pope *ad experimentum*.[26] The Section for Migrants and Refugees proposed 20 action points as specific ways to implement aid solutions, both in policy and in the attitude and action of Christian communities.[27] The influx of huge numbers of refugees from Ukraine was becoming a challenge for both states and local communities, local government units, and church administrations, dioceses, and parishes at a very rapid pace. This fact has undoubtedly caused an upsurge in various institutions and environments, provoked the mobilization of initiatives, and triggered the necessary relief efforts. Despite the element of spontaneity, these needed to be organized and coordinated. It verified the degree of reception of the Pope's teaching and the living out of the faith, the sense of community in both the religious dimension and the building of civil society.

## 4. Methodological Findings

In order to identify the motivation behind the activity of the parishioners and their spontaneous involvement in helping refugees and to confront it with the social teaching of the Catholic Church, the examples of the parish of St. Joseph in Chorzów in the Silesian Voivodship in Poland and the parish of the Holy Spirit in Kátlovce in Slovakia[28] were used. In this respect, it is important to briefly grasp the historical and social context of the parishes studied. St. Joseph's parish originated from St. Barbara's parish in Krolewska Huta and was separated from it in 1912. The church was dedicated in honor of God and its patron, St. Joseph, Protector and Advocate of the Church. The new church was consecrated by the Revd. Dean Wiktor Schmidt from Katowice on 18 November 1907. In its more than 100-year history, St. Joseph's Parish in Chorzów has had only four parish priests (which is a peculiar phenomenon), and to date it has had 61 vicars. At present, it has about 11,500 parishioners.[29] Its pastor is Dr. Rafał Śpiewak, co-author of this article. This allowed direct observation of the case described.

The earliest known written reference to the village of Kátlovce is connected with a land register from 1401, where the name Katlowicz appears.[30] In a reference from 1405, the name Kátlovce is given as Kathloch, and later in the century, another German form, Kathlowicz, and the Hungarian form, Kathló, are also known. At this time, Kátlovce is mentioned in written sources under various names.[31] The local parish church dates from 1720. It was built in the Baroque style.[32] The parish has a rural character.

The following methods were used in the research process: literature research, case studies, the social sciences, and the Individual In-Depth Interview (IDI) method. The term case study is synonymous with the monographic method and English-language terms such as *case study* or *case study method*. A case study is defined as a research project in which the object of study is usually a single case (Pizło 2019). The research process (Individual In-Depth Interview—IDI) was based on a pre-drawn interview scenario. All the questions contained therein were open-ended. The interview technique adopted here allowed comparisons to be made in the research material obtained (Dźwigoł 2018, p. 163). Interviews were conducted between July and September, and the selection of experts was arbitrary, which is a characteristic of the methodology used (Meuser and Nagel 2009, pp. 17–24).

Interviews were conducted with selected participants in refugee assistance projects in both Polish and Slovak parishes. The international cooperation of the researchers made it possible to make comparisons and identify analogies in the behavior of the parishioners. The number of respondents in both countries was similar. In the Polish parish, the study included 32 respondents, and in the Slovakian parish, 29 respondents. The criterion for selection was the involvement of participants in parish projects to help refugees. An attempt was made to obtain statements from a diverse group of participants (age, material situation, involvement in church life, etc.). The case study method was based on the implementation of the research process in individual, planned, and consequent stages. In the first stage, the themes and objectives of the *case study were* established. The questions were: do the actions of the parishioners result from their understanding of the social teaching of the church in the context of helping refugees or migrants? How were the motives for their actions and the results obtained introduced by their identification with the Catholic Church and its social teaching? In stage two, the study area was defined—the parishes of St. Joseph in Chorzow and St. Spirit in Kátlovce. They are typical, traditional parishes of the Roman Catholic Church. The Polish parish has a diverse social profile, while the Slovakian one has a homogeneous profile. At the next stage, the researchers familiarized themselves with the conditions under which the parishes function in the context of the Polish and Slovak churches. Subsequently, parish statistics and descriptions of similar activities undertaken in the past were studied. An important stage was to interview selected parishioners organized into a support group. In the final stage, the collected material was sorted in terms of its relevance for capturing the investigated forfeit. The seventh stage proceeded to describe

the *case study in relation* to its originally assumed cognitive function in the context of the Church-specific social situation.

The Polish parish respondents included 55% women and 45% men. A total of 8% were under 25 years of age, 26% were between 25 and 40 years of age, 21% were between 40 and 60 years of age, and the remainder were over 60 years of age. A total of 38% declared a very good financial situation; the rest described it as average. In the Polish parish, 82% of respondents declared that they belonged to a Catholic church. In the Slovak parish, 63% of women and 37% of men took part in the survey. A total of 6% were under 25 years old, 19% were between 25 and 40 years old, 32% were between 40 and 60 years old, and the rest were over 60 years old. A total of 23% declared a very good financial situation; the rest described it as average. All respondents declared that they belonged to the community of believers in the Catholic Church.

The semi-structured interview schedule included questions common to both groups of respondents:

1. What were the main motives for your participation in the project to help refugees from Ukraine?
2. Did you have relationships with people from Ukraine before the war broke out? Which ones?
3. Have you so far been involved in actions to help (volunteer) refugees?
4. In which activities did you participate in the parish project to help refugees from Ukraine?
5. What were the greatest difficulties?
6. To what extent does this project affect the image of the Catholic Church (parish) in the local environment?
7. How has the project been disseminated in your community?
8. Was the fact that you are a Catholic parishioner relevant to your involvement in helping Ukrainian refugees?

## 5. IDI Analysis Conducted as Part of the Research Process

Respondents were asked to respond to the issues mentioned above. It should be noted that in the course of the interviews, all respondents, both in the Polish and Slovak parishes, declared that their answers were sincere. In the course of the interviews, efforts were made to be as honest as possible and to avoid any suggestions. For the analysis of the collected material, in order to organize the statements obtained, a key related to the issues contained in the individual questions was adopted. The examples cited in the analysis illustrate the most frequent responses, which are similar in meaning.

## 6. Motivations for Helping Refugees

From the interviews obtained in the Polish parish, the nature of the motivations for joining a group to help refugees from Ukraine is very diverse. Responses indicated several motives for aid activity. The first motive that the participants were guided by was the pattern of behavior brought up in the family home. This reasoning appeared in the statements of 65% of the respondents. The responses were very similar but also varied due to the different experiences of the respondents. They are illustrated by the following statements: "The main motivation for my participation was the values passed on by my parents. There was always an atmosphere of helping those in need in our home". Another motive was the need to repay the debt for the help received from other people in the past. This is represented by the following statements: "We ourselves experienced financial problems in the past, when help for our family flowed from abroad" and "Part of our family emigrated to West Germany and when we had financial problems we could count on their help".[33]

A part of the respondents, i.e., 53%, emphasized the role of the direct influence of the parish priest and the rhetoric he used: "I was motivated by the invitation of the parish priest and by the awareness of the great danger that can affect any of us in the absence of

interpersonal solidarity". A key motivation vis-à-vis the stated hypothesis of the research was that the respondents indicated a link between their activity and the social teaching of the Church. There was an overwhelming number of such indications (over 60%). This is illustrated by the following statements: "I go to church and I can't imagine not helping in a situation such as war. I am a Catholic and what I believe determines my everyday life". Another motivation worth noting was the teaching of Pope Francis: 'I was encouraged by the teaching of Pope Francis and his appeals to help refugees. He said it with conviction, citing the Gospel".

There were also responses in which the motive for involvement was to seal or reaffirm a professed faith. "My whole family participates in the life of our parish. If we didn't help I would consider that what is said in the sermons doesn't make the slightest sense"; "I help because I do it for Christ"; "What is there to talk about, every good deed comes back to a person. I believe in it and for that I help and will continue to help". In almost all responses, the religious aspect dominated as an important motive for involvement in helping refugees. The professed faith thus found its confirmation in life in the face of a specific challenge. The content of the teachings of both Pope Francis and local pastors provided a valuable impetus for the commitment to help.

In the Slovak parish, the main motives of the selected respondents stemmed from their family backgrounds. More than 55% of respondents said that their parents' upbringing and example played a key role. In second place, more than 25% of respondents stated that they themselves had needed help in the past and, on that basis, did not hesitate to help those in need. Some respondents expressed that they were also directly influenced by the parish and the local parish priest, who kept parishioners informed about events in Ukraine. This was particularly reflected, for example, in material aid, where residents collected clothes, food, and hygiene items in the parish. On this occasion, with the participation of the parish priest, the course of events in Ukraine was discussed on an ongoing basis, the current situation was assessed, and consultations with the parishioners were undertaken as a result. An additional motivating factor for respondents was the fact that Pope Francis also called on the parishioners to help the Ukrainians. A large part of the responses regarding the motives for helping were dominated by the factor of religious feelings and the mutual example of co-religionists.

## 7. Relations with Ukrainians before the Outbreak of War

Another question concerning Polish parishioners was the identification of familiarity with Ukrainians by those engaging in outreach. The answers varied, but the overwhelming majority of people surveyed, around 60%, had no previous relationship with Ukrainians. In extended questions beyond the strictly worded original question, it appeared that the fact that it was Ukrainians and not another nation did not matter to the helpers. Examples of statements: "Part of my family comes from the territory of today's Ukraine"; "I often met in shops, service points, on the street when they came to our city. Previously, I knew one family with whom my relative kept in touch"; "I had no contact with Ukrainians. In the past [during the times of the USSR, author's addition], even when I went to school, I associated them rather with Russians. I was not interested in history. They are people to me and I would help if another nation found itself in such a dramatic situation"; "My only information about their situation came from the media, I had no direct contacts. The media showed these people, in different situations, every day. It made a very depressing impression on me, a terrible tragedy."; "My only contact with people from across the eastern border was during trips abroad on holiday. The Ukrainians seemed close and we made contact easily. But I have to admit that I didn't really distinguish between them and the Russians, because they spoke Russian. It was only with the Orange Revolution that I realised they were two different nations. Knowing the Ukrainians directly didn't matter much to me"; "I had contact with people working in our region, usually in trade or construction. They had a reputation as good workers".

It should be emphasized that previous knowledge of Ukrainians was of little importance to aid workers (nearly 75%). There were no family ties or the maintenance of friendships. From the content of the statements, one could receive the impression that assistance to Ukrainians was essentially humanitarian in motive and not based on national proximity, historical, or interstate ties. In several statements, there was a historical theme concerning the period when Ukraine belonged to the Soviet Union. An interesting theme was the difficulty of distinguishing between Ukrainian and Russian nationalities on the grounds that some Ukrainians speak Russian on a daily basis.[34]

In Slovakia, as many as 72% of respondents had no previous relations with people from Ukraine but kept up to date with the current situation. 10% of respondents stated that they have contacts with people from Ukraine that they acquired during holiday trips. 2% of respondents indicated that members of their family live in Ukraine. It is worth mentioning that the parish of Kátlovce is located near the village of Jaslovske Bohunice, where the Bohunice nuclear power plant is located, whose construction involved Ukrainians in addition to Russians. It follows that the respondents had contact with Ukrainian residents. Currently, there are also many Ukrainians on the labor market in Slovakia in various sectors. It can be inferred from the interviews that the assistance was not provided on the basis of national proximity or other connections. It was assistance that the Ukrainian refugees needed, undertaken on humanitarian and religious grounds.

## 8. Experience in Charity Assistance

It should be noted that more than 50% of the respondents had not previously been involved in organized help activities. They usually associated help with temporary help received from or given to other family members, friends, or neighbors. However, the vast majority had been involved in giving money at parish collections (over 90% of respondents) or secular charity collections. Their responses were as follows: "Every year I supported the Great Orchestra of Christmas Charity campaign. The money from this huge action has been supporting the Polish health service for years. I was active with my wife in the local committee for many years"; "I participated in financial collections. I wasn't directly involved in any action, but I tried to donate money to various noble targets."; "I didn't actually support something specifically. But the needs are many and it's worth it if you have the opportunity to help others. It doesn't cost much"; "I have always financially supported numerous parish initiatives and those announced by Caritas. For me this is the best form of support for those in need. The people who organise the aid know where to put the money"; "I have been involved in aid campaigns at the school for years. I am a teacher and it is my duty to set an example for my pupils and indirectly for their parents. Not enough is said about the fact that help is necessary and in a tragedy such as the war in Ukraine, it is absolutely necessary to help"; "So far, I have not been directly involved in the activities of any charity, charitable organisation. I cannot say why. Why am I doing it now? I watch TV every day, I see what's going on and I can't believe it. It's only when these people came to our town that I see that it's all true. This has mobilised me. After all, it could have happened to us too".

From the statements obtained, it appears that those involved did not have much organizational experience of outreach before. This indicates that, in general, their involvement was spontaneous. It should be emphasized that this was influenced both by direct experience of the presence of Ukrainians arriving in Poland and by the content of media messages. The latter sometimes gave rise to a sense of distance and disbelief. This is particularly confirmed by the last statement quoted. Especially in the first three months of the war, all television stations were filled daily with content depicting the situation in Ukraine, which was dominated by messages emphasizing the drama of the war.[35]

Forty-two percent of Slovak respondents said that they were involved in various charitable volunteer actions. Charity collections were carried out in two directions for Ukrainian residents—one direction concerned Ukrainians who left Ukraine before the threat of war and temporarily stayed on our territory. The second direction of charity was

oriented towards Ukrainians who remained on the occupied territory and needed help. The majority of respondents were not only willing to help in the form of food and clothing collections but were also ready to provide temporary accommodation for Ukrainians. The refugees were mostly women with young children. Even members of one family of refugees from Ukraine were considering applying for asylum and permanent residence in Slovakia because they had found employment here. Responses to this question confirmed that respondents had previous experience with activities to help people in difficult life situations. Their actions were also influenced by media reports, which regularly reported on the suffering of the Ukrainian population as well as the drama of the war.

## 9. Areas of Refugee Assistance

Assistance to refugees was organized in the Polish parish in a number of areas. These were determined by the emerging needs and the social structure of the refugees. It can be said that the dimension of material assistance was the most significant, and the most important was help in finding accommodation. An important aspect was mediating with local government institutions due to their lack of knowledge of the Polish language and lack of medical and educational assistance. This is illustrated by the following examples of statements: "I was the organiser of accommodation for the Ukrainians. Especially in the first days, help in preparing housing was absolutely necessary. We had trouble finding places to stay. Sometimes this required small renovations to the flats. We found money for this and everyone was very committed. The Ukrainians, especially the mothers, were very grateful. It was the best payment for us". "I made my own house free and available. To be honest, it took very little time for me to decide like that. The drama they showed on the media was a unique and unconditional motivation to help. I will never forget the expression on the faces of the people coming from Ukraine, their sadness and visible horror". "I participated with the collection of food and hygiene items. It went very smoothly. Many of our parishioners brought the necessary articles. In general, we had no problems with the collections, especially in the first two months". "People responded spontaneously, bringing products, including many food products. We prepared parcels for Ukrainians according to specific needs. Basically, every day there was something to do. I have never been in such a situation, it's not the same as helping the poor or the sick. The fact that these are people driven out of the country by the thugs from Russia is unthinkable". "I was involved in helping with official matters, registration of Ukrainians at the City Hall. I studied Ukrainian and Russian and when I heard in church that such help was needed I immediately volunteered". "I take part in helping to arrange a school for the children. I am the headmistress of the school and these issues are closest to me. I also have a lot of friends in schools and that helps a lot". "I am a nurse and I joined in to help provide medical care for the baby and his mother. I was able to help them quickly". "Wondering what I can be useful in I thought, because of my knowledge of many entrepreneurs I help to arrange jobs for people who declare their willingness to do so. Contrary to what you might think, there aren't many of them, but I understand that a difficult situation makes important decisions difficult". "I organised integration trips"; "I was involved in helping to organise the parish fete, during which Ukrainians sold handmade products. They lived through it a lot, they cared about appearing as good as possible"; "I helped with the collection of clothes. We did it twice a week in church. People from our parish brought the things they didn't need willingly, they didn't need to be persuaded". From the statements quoted, the scale and multifaceted nature of the aid organized take shape. Direct action, the almost all-day commitment of the coordinators and participants of the aid group, was predominant. In addition, the support of other parishioners played a great role, who, although limited to bringing gifts and donating money, nevertheless proved crucial in the face of emerging material needs.

The parish project to help refugees from Ukraine focused on the Kátlovce parish and throughout Slovakia. The parish organized assistance for those Ukrainians who found temporary shelter in the homes of its residents. This involved serving meals throughout

the day at the local primary school. Conditions were created for pre-school children to attend school, and similarly for school-age children. A Slovak language course for beginners was also organized at the vicarage. The parish also provided specially prepared accommodations. Assistance within the parish project manifested in two members of the parish taking part in humanitarian work in special centers for refugees on the border with Ukraine. One member of the parish drove a bus that distributed refugees from Ukraine throughout Slovakia.

Summarizing the various initiatives, it can be said that the assistance was multifaceted. Without hesitation, people were ready to sacrifice a lot of their personal belongings for the refugees from Ukraine. They were able to coordinate very quickly for the benefit of those in need. The following theme emerged in people's statements: "Each of us did whatever our material and housing conditions allowed us to do"; "We devoted all our free time to helping the refugees. We gave them clothes, hygiene products, handed over small money directly. We have not been so involved in anything for years".

## 10. Difficulties of Project Implementation

The researchers asked the participants of the refugee assistance project in Chorzów what caused the greatest problems for this group. According to their statements, the biggest limitation was the Ukrainians' lack of knowledge of the Polish language. Added to this was the difficulty of communicating in other foreign languages, especially English. The exception was Russian, which was spoken by the majority of refugees, with a concomitant lack of knowledge of Ukrainian. This aspect of the relationship required translators. Mutual contact was facilitated by the admittedly limited knowledge of Russian among the elderly. Of the key issues, due to the large number of people arriving, serious difficulties were encountered in finding accommodation. Sometimes reported accommodations needed to be revitalized and prepared, which generated additional costs and the need for organizational activities in this area. Uncertainty about the timing of the provision of private accommodation to refugees was also a problem. This circumstance required precise arrangements and the securing of relief measures directed at those who made their dwellings available in the event that difficulties arose in maintaining the refugees. "We could not predict for how long housing and financial assistance would be needed. It was difficult to plan the budget also due to the uncertainty of the support received". A major problem, according to the Slovak parish respondents, was the language barrier. Especially among the younger support participants, some older people still remembered Russian and were able to communicate with Ukrainians. Additionally, for this reason, opportunities were created to run a language course at the vicarage. A total of 46% of the respondents mentioned the alleged length of stay of the Ukrainians in the parish as a major problem, as it was not possible to estimate in time the accommodation and food needed. Another problem that arose during the parish *outreach* project was that parishioners who participated in the outreach were mostly willing to donate food, clothing, and hygiene items but not cash. Based on a review of individual respondents' answers, the biggest problem was that the costs incurred could not be included in the project budget, creating the possibility of applying for *match funding* for the project.

## 11. The Importance of the Project for the Image of the Parish

Another issue was to find an answer to how, in the opinion of the participants of the outreach, the activities carried out in the parish affect the image of the Catholic Church (parish) in the local environment? The researchers considered that the size and importance of the campaign were so spectacular that it should have a positive impact on relations between parishioners and, consequently, also on the image of the Church. This assumption was verified positively, as illustrated by the following selected statements of the respondents: "In my opinion it has enlivened the life of the parish. There has been greater integration within the parishioner community".

"The commitment of the group of people who carried out so many activities found a positive reception from many parishioners, whose presence is visible in the church, who donated funds to the goal of comprehensively assisting Ukrainians". "Parishioners reacted very kindly to our activities. This was particularly evident during the parish fete, during which Ukrainian women sold their culinary products and the money raised enriched the aid fund". "In conversations in many families, the topic of our parish's help for Ukrainians was often present. I know this from conversations with many people, with whom we always meet briefly after Mass in front of the church". "Our parish stood out from other parishes in the city in this respect. The fact that it was visible influences people to think better about the Church, which in recent times has been going through, in my opinion, a crisis. Especially a crisis of its image". The project carried out in the parish of the Holy Spirit in Slovakia has significantly influenced the outlook of the Catholic Church in the community. The importance of the action lies not only in helping the people from Ukraine, but it also served to change the mentality of the people in the parish. It increased mutual bonds and positive relationships. Respondents indicated that people's involvement in charity events and fellowship among parishioners increased. They were able to improve communication among themselves and identify the importance of priorities in the implementation of the project. The coordination skills of the parish priest, who has the ability to motivate his parishioners and emphasize the importance of helping those in need, also played an important role. The implementation of the project has changed the mentality of the people in the region regarding the activities and deeds organized by the church. Additionally, on the basis of this activity, it can be assumed that parishioners are ready and willing to help in any other charitable activity.

## 12. Forms of Promoting the Project

The formation of opinions about the involvement of the parish faithful in helping refugees was essentially built on direct relationships between people. Another aspect was the deliberate use of parish media. The size of the campaign and its forms also became the focus of external media attention, as pointed out by interview participants. The following selected statements illustrate this: "Information about the campaign, comments from parishioners, were in our parish newspaper at all times. It is available to all interested parties and has been not only popular but also enjoyed by our community for many years". "The quickest and easiest way to read about us and the Ukrainians was by going to our parish website. The information was kept up to date and, as far as I know, was of great interest, not only to our parishioners but also to others not living in the parish area". "I have learned that our outreach has found interest on the town's social media. It's good that this has happened, because I hope it has strengthened the commitment of Chorzowians to help Ukrainians. It showed that it is not only possible, but also very much needed at this difficult time for them". "About the initiative there was a piece of journalistic material prepared and aired on the state TV news programme," he added. "We watched with friends a programme on state television about what we were doing, about our assistance. I must admit that it gave us a lot of joy and satisfaction. I had never been on television and for a tiny part of the programme I was visible".

A range of available communication tools were used to promote the project in the Slovakian parish. These were used in both online and offline formats. For the majority of parishioners interviewed, information was communicated through the parish priest at Masses and other face-to-face meetings of parishioners on the parish premises. Participants at Masses and other meetings gradually passed on information about the project to their family members, friends, and neighbors. The main communication point was the parish environment. Respondents also received information through the local radio station as well as the community website, which contains information intended for parishioners.

After evaluating the individual responses, it can be concluded that the communication system set up in the implemented project focusing on helping Ukrainians was of the

required level and fulfilled all the objectives that were set for it. It was apparent that the local parish priest had sufficient experience implementing charitable projects.

### 13. The Importance of Religious Inspiration

A key research question, related to the first question, was to find out to what extent the fact that the helpers were Catholics made a difference to the activity they undertook. The relationship between people's attitudes and their values is an interesting area of research. This justifies the need for many institutions to build social relations on an ethical basis with a high degree of openness towards the other person. Among others, the following statements were obtained: "I think it was a reflex of the heart. Of course a faith-based upbringing was of great importance for me and my sensitivity. This has always been the case and therefore I don't think I'm doing anything extraordinary, this is how every Catholic should behave". "Helping a neighbour in need is a test of faith," he added. "Living the gospel has been a fundamental motive for my commitment and help. Probably for this, too, I have the greater satisfaction of being able to do it selflessly, without any material benefit to myself"; "I have never experienced help from others and therefore I was critical of such indifference. I decided to take a different attitude and I am proud of it. It is a pity that people pay less and less attention to each other on a daily basis and help each other even less"; "Solidarity with the suffering and vulnerable is for me the main place to meet God"; "It helped me to deepen my faith and to be in the Church"; "The most important influence for me was the religious upbringing in the family home. My parents were always deeply religious and were always open to other fellow human beings"; "I did not act out of religious motives, but it was a human reflex and a need of the heart. It doesn't give it any special meaning. Everyone should help in this situation".

All of the participants in the study at Holy Spirit Parish are from Catholic backgrounds. They expressed the opinion that every believing Catholic should be characterized by a special sensitivity to help their neighbor. Therefore, in almost all responses (93%), there was a belief that helping those in need is not unusual. It was described as normal behavior for any Catholic. Some respondents emphasized that helping those in need is a test of faith, citing knowledge of the Gospel requirements. None of the interviewees mentioned any kind of payment for the help they provided. All of the activities undertaken were non-profit, as otherwise, they stated, it would be contrary to the religious motive. The Slovak pastor indicated that faith is deeply rooted in his parish, and the situation studied reflects and confirms its depth and authenticity.

### 14. Discussion

In carrying out this research project, the researchers agreed with the assumption described in the scientific literature that both quantitative and qualitative research play an important role in science. They assumed that the effect of the research conducted using this method would be conclusions of a descriptive nature, allowing for the verification of the correctness of the existing description of the studied phenomenon or the formulation, in accordance with the research assumptions, of proposals for new original conclusions. The research process (Individual In-Depth Interview—IDI) was based on the preparation of an interview scenario. The semi-structured interview technique adopted allowed comparisons to be made in the research material obtained. The originally assumed issues during the research were expanded with additional information from the respondents. The interviews were conducted substantively, in a good atmosphere, without suggesting answers, and with a high degree of freedom for the respondents. Conducting the interviews in a face-to-face manner also allowed for observations of the respondents' behavior. This important cognitive aspect facilitated the formulation of follow-up questions.

Based on the material collected, the underlying motives for involvement were identified. Out of a total of 32 people in the Polish parish, the vast majority were believers and practitioners. The same applies to the Slovak parish. In both cases, this fact had a fundamental influence on their involvement in helping refugees from Ukraine, as the

results of the interviews proved. In their statements, the respondents emphasized the fact of belonging to the Catholic Church, the connection of their motivations to their faith, and in particular, the social teaching of the Catholic Church. The teachings of Pope Francis were cited most often, although this was not a clearly distinguishable trend. In addition, the variety of ongoing forms of outreach that were undertaken by parishioners was identified. The main difficulties were also identified, the biggest of which was the problem of communication due to the lack of knowledge of Polish and Slovak. In-kind and financial support ran smoothly as a result of the wide response to parishioners' appeals. A major problem was finding accommodation for the refugees in the longer term, such as for a few days. This was solved mainly through individual invitations to the private homes of parishioners. Participants in the aid groups assisted Ukrainians in contacting local government institutions with the necessary formalities. In addition, they provided assistance in finding work to those who declared a need for it. The aid groups enjoyed the understanding and support of the parish community. They also positively influenced the establishment of closer relations and even friendships, which may result in breaking stereotypes burdened by historical experiences in the future. Especially in Polish-Ukrainian relations.

It is worth noting that all aid activities were facilitated by the fact that they were disseminated both within the parish communities and externally through local and, in the case of Poland, national media. For the internal communication of the team of people involved in helping refugees, special groups were created on WhatsApp.

After the research process and the analysis of the collected material, it must be concluded that popularized evangelical values were to a large extent the driving force behind the involvement in aid projects in both the Polish and Slovak parishes. The activities undertaken enlivened the life of the parish and brought parishioners together, and a significant number of them joined in the support activities, especially the financial support of refugees. Without their support in kind and financially, the assistance would have been far less significant. According to the respondents, the aid actions had an impact on the image of the parish in the parish community and its surroundings. These actions aroused interest, all the more so because there was less attention paid to the local dimension of aid in the national media coverage. Illustrating the help of thousands of people, without whom the plight of refugees would be much more difficult. Consequently, according to the respondents, these initiatives influenced the creation of a positive image of the institution of the Church itself in society.

As a result of the project, several conclusions of a pastoral nature emerge:

- Due to the current social context, the topic of the social teaching of the Church should be addressed to a greater extent during Sunday homilies;
- Groups operating in the parish should diagnose the most urgent social challenges and seek practical forms of implementing the commandment to love neighbors;
- Professional training of group leaders in management, marketing, and media is needed to increase parish initiatives;
- Since charitable projects not only engage and integrate the parish community, but they should also be professionally promoted in the media due to their positive impact on the image of the Church;
- It also seems beneficial to build platforms for the exchange of experiences between individual parishes and to take stock of the projects undertaken.

Catholic social teaching is a synthesis of the Gospel in the face of the current challenges of social life. The role of the Church is to undertake ongoing reflection on the dynamics of social life so that the documents produced include, on the one hand, assessments of the situation and, on the other, formulated directives for action. In the project carried out, the assumption was confirmed that church documents create a space for the parish to respond to crisis situations. The behavior of the faithful initiated in the parish communities was inspired by the Church's teaching, in particular that of Pope Francis on refugees.

In both parishes surveyed, the process of motivating the faithful to aid activities was diagnosed by referring to their professed values. On the other hand, the pastoral condition

of these parishes made it possible to design and carry out, in a short time, an effective aid process in the face of the refugee crisis related to the war in Ukraine.

The effect of helping the Ukrainian refugees was to revitalize parish life, consolidate parishioners, and make them more sensitive to the needs of others. An important aspect of the measures taken was also to bring the representatives of the different peoples and Churches (Catholic and Orthodox) closer together. Taking into account the benefits of the refugees who arrived in the area of the parishes under study, in the short term, the measures taken provided them with comprehensive assistance in meeting their basic living needs. In the long term, they created conditions for the process of adaptation and integration in the new place of residence, in particular health care, education, and finding their way in the local labor market. As a result of the measures taken by the two parishes, there was an increased sense of security among the Ukrainian refugees, which was crucial given the wider context of the crisis. The social situations generated by ongoing processes in the world are extremely diverse. In this regard, it should be pointed out that, in order to be properly situated in them, the general indications of the Church's social teaching may not suffice. Hence, in specific situations, there is a need to prioritize and create models appropriate to the realization of the real human good. The activity undertaken in the parishes surveyed fits in with the above assertions.

## 15. Completion

From the first day of the war, Polish and Slovak society provided selfless assistance to Ukraine. One of its cornerstones was the acceptance of a wave of refugees, unprecedented in the realities of the 21st century, leaving their country in extremely dramatic conditions, often without means of subsistence or prospects of return. Its scale, with people arriving in their thousands, was hitherto unknown in the post-war histories of Poland and Slovakia and more widely in other European Union countries that experienced the effects of the Second World War. The enormity of the material devastation and the millions of human tragedies, and the sensitivity to the fate of the victims of the war dramatically waged across the eastern border, stimulated the communities of the Catholic Church to engage in comprehensive relief efforts. The dimension of Poles' activity surprised themselves, the country's political authorities, and the international community.[36] Poland became a leader in international aid, gaining the gratitude of the Ukrainian authorities and its citizens. This gratitude was shown by refugees in their contacts with Poles who provided them with assistance. Helping refugees became a priority for the government of the Republic of Poland and for hundreds of NGOs and social organizations working in this area. Additionally, for the Slovak authorities and the people there, helping refugees from Ukraine became not only a political but also a social priority. The teaching of the Catholic Church was crucial in organizing Polish and Slovak assistance for refugees from Ukraine. It is important to emphasize that the decisions to involve the two parishes under study in the relief effort were taken immediately after the outbreak of war, and the scale and forms of assistance were extremely wide. In both countries, the Church initiated a huge involvement of the parishioners gathered in parishes throughout the country. The relief activities took place on the basis of the social teaching of the Church, materializing Christian values on a social and individual Catholic scale.

The present research is part of the ongoing synod on synodality in the Church from 2021, which seeks, among other things, answers to the question of the role and tasks of the laity in the evangelizing mission of the Church.[37] In the current reality, it is important to strive to find an appropriate form for the synodal Church and to identify theological principles and precise directions for pastoral action.[38] The realization of the synodal Church is a necessary prerequisite for a new missionary impulse that will involve the whole people of God. In this context, the exposure of concrete initiatives of both a religious and a social nature, which are the result of cooperation between clergy and laity in the parish, can give an ascent to the identification of the plans of pastoral mission in the current social context. The experiences described here have cognitive value but also inspire the design of further

evangelization activities, in which the most important argument is the living witness of faith that verifies itself in the face of existential challenges. Synodality is lived out in the Church for the strengthening of its missionality. The overarching mission of the Church is evangelization (Paul 1975). The whole people of God is the object of the proclamation of the Gospel (Second Vatican Council 1967). In it, every baptized person is called to be a promoter of mission because we are all missionaries. The Church is called, in synodal synergy, to activate the ministries and charisms present in her life and to listen to the voice of the Holy Spirit in order to discern ways of evangelization. According to this, the Church, insofar as she is Catholic, makes universal what is local and local what is universal. The particularity of the Church in one place is fulfilled in the heart of the universal Church, and the universal Church is revealed and realized in the local Churches. Hence, the actions taken by individuals belonging to the community of the Church, inspired by it and in communion with it, implementing the fundamental Gospel message of fraternal love and love of neighbor, affect the image of the Church in public perception and the effectiveness of witness.

The research process undertaken was carried out on the Polish side by researchers from the University of Economics in Katowice and the University of Silesia in Katowice, and on the Slovakian side by researchers from Cyril and Methodius University in Trnava. It was based on inquiries into the form in which the aid was provided and, especially, the motives behind it. It should be emphasized that both societies are largely linked to the Catholic Church. From this phenomenon arose the researchers' finding that it would be valuable to compare the behavior of parishioners. The main task was to verify the research thesis that the influence of the social teaching of the Catholic Church on the parishioners in the area of the Polish and Slovak parishes selected for the analysis of the described phenomenon was crucial. This thesis was verified positively. The difficult experiences of parishioners and their humane, selfless attitude towards people unknown to them confirm the necessity of disseminating the values that have been promoted by the Catholic Church for years and spectacularly done by Pope Francis in recent years (Widera 2020).

**Author Contributions:** R.Ś.: Conceptualization; Formalanalysis; Investigation; W.W.: Investigation; D.J.: Project administration; T.J.: Resources. All authors have read and agreed to the published version of the manuscript.

**Funding:** The research was funded by the statutory research of the authors' individual universities.

**Informed Consent Statement:** Informed consent was obtained from all participants taking part in the study.

**Conflicts of Interest:** The authors declare no conflict of interest.

## Notes

1.  Official Twitter profile of the Border Guard and article: https://300gospodarka.pl/news/uchodzcy-z-ukrainy-w-polsce-liczba (accessed on 1 November 2022).
2.  https://www.gazetaprawna.pl/wiadomosci/swiat/artykuly/8423491,onz-liczba-uchodzcow-przekroczyla-100-mln-wojna-w-ukrainie.html (accessed on 2 May 2022).
3.  See note 1 above.
4.  Official Twitter profile of the Border Guard and article: https://300gospodarka.pl/news/uchodzcy-z-ukrainy-w-polsce-liczba (accessed on 10 March 2022).
5.  The government has adopted a special law on aid for refugees from Ukraine: https://www.gov.pl/web/mswia/rzad-przyjal-specustawe-dotyczaca-pomocy-dla-uchodzcow-z-ukrainy (accessed on 10 April 2020).
6.  https://www.dw.com/pl/s%C5%82owacja-pomaga-ukrai%C5%84skim-uchod%C5%BAcom-wsparcie-tak%C5%BCe-dla-os%C3%B3b-przyjmuj%C4%85cych-ukrai%C5%84c%C3%B3w/a-60932555 (dostep 10 April 2021).
7.  https://ua.gov.sk/sk.html (accessed on 22 October 2022).
8.  https://www-employment-gov-sk.translate.goog/sk/uvodna-stranka/informacie-odidencov-z-ukrajiny/informacie-odidencov-z-ukrajiny.html?_x_tr_sl=sk&_x_tr_tl=pl&_x_tr_hl=pl&_x_tr_pto=sc (accessed on 10 October 2022).

9 Guide for Hosting Refugees from Ukraine https://odpowiedzialnybiznes.pl/wp-content/uploads/2022/02/Poradnik-dla-osob-goszczacych-uchodzcow-z-Ukrainy.pdf (accessed on 11 March 2022).

10 Cf.: https://www.ekai.pl/bp-smigiel-kosciol-w-polsce-zrobi-wszystko-by-pomoc-ukraincom/; https://www.ekai.pl/abp-guzdek-dziekuje-wiernym-i-ksiezom-za-pomoc-ukraincom/; https://www.ekai.pl/ludzie-kosciola-o-tym-jak-madrze-pomoc-ukraincom-dlugofalowo/; https://www.ekai.pl/?s=Pomoc+Ukrai%C5%84com (accessed on 3 November 2022).

11 https://caritas.pl/ukraina/ (accessed on 10 October 2022).

12 https://episkopat.pl/ (accessed on 10 October 2022).

13 https://archidiecezjakatowicka.pl/o-diecezji/aktualnosci/2541-caritas-to-sily-szybkiego-reagowania-kosciola-2 (accessed on 16 September 2022).

14 https://diecezja.pl/aktualnosci/prawie-900-miejsc-noclegowych-dla-uchodzcow/ (accessed on 28 February 2022).

15 Klauzinski S.: The church helps refugees from Ukraine? https://oko.press/kosciol-pomoc-biskupi-wojna-w-ukrainie/ (accessed on 12 April 2022); Jakubowka D.: How the Kraków church helps refugees https://lifeinkrakow.pl/w-miescie/4335,jak-krakowski-kosciol-pomaga-uchodzcom-nikomu-nie-odmawiamy-wsparcia. (accessed on 12 April 2022); Borek Z.: Churches with help for refugees? A sin not to use them https://www.polityka.pl/tygodnikpolityka/spoleczenstwo/2158089,1,koscioly-z-pomoca-dla-uchodzcow-grzech-ich-nie-wykorzystac.read (accessed on 12 April 2022).

16 https://utecenci-kbs-sk.translate.goog/?_x_tr_sl=sk&_x_tr_tl=pl&_x_tr_hl=pl&_x_tr_pto=sc (accessed on 11 October 2022).

17 https://www.vaticannews.va/sk/cirkev/news/2022-02/cirkev-na-slovensku-sa-mobilizuje-v-pomoci-ukrajinskym-utecencom.html (accessed on 11 October 2022).

18 See Note 17.

19 https://www.abu.sk/archiv/spravy/trnavska-arcidiecezna-charita-poskytla-potravinovu-pomoc-pre-viac-ako-dvadsat-ludi (accessed on 3 November 2022).

20 https://www-abubask.translate.goog/abuba/helpukrajine?_x_tr_sl=sk&_x_tr_tl=en&_x_tr_hl=en&_x_tr_pto=sc (accessed on 12 October 2021).

21 https://archidiecezjakatowicka.pl/ (accessed on 3 November 2022).

22 https://www.abu.sk/ (accessed on 3 November 2022).

23 In a broader sense: Zimny (2006).

24 https://www.ekai.pl/papiez-parafie-to-nie-kluby-dla-nielicznych/ (accessed on 31 January 2023).

25 https://www.vatican.va/roman_curia/congregations/cfaith/documents/rc_con_cfaith_doc_19860322_freedom-liberation_pl.html (accessed on 24 October 2022).

26 On 23 April 2022, Cardinal Michael Czerny became prefect of the dicastery. The precedent is that the Section for Refugees and Migrants is personally headed by Pope Francis.

27 https://misericors.org/20-punktow-dzialalnosci-ws-migrantow-i-uchodzcow-dykasteria-ds-integralnego-rozwoju-czlowieka/ (accessed on 30 August 2022).

28 https://www.dokostola.sk/kostol/411863-svateho-ducha (accessed on 20 October 2022).

29 Roman Catholic Parish of St Joseph in Chorzów https://www.sw-jozef-chorzow.pl/historia.html (accessed on 10 April 2022).

30 http://www.katlovce.sk/ (accessed on 20 October 2022).

31 Interesting connections can be found between the geographical relief of the district of Kátlovce and the name of the village. The village lies along the Blava stream in a kind of basin. In the Hungarian-Slovak dictionary the word katlan appears in this context—which corresponds to the Slovak equivalent - kotol, krater. A basin, geographically speaking, is defined as a closed depression of various origins, with a surface significantly softer than the surrounding terrain.

32 https://www.pamiatkynaslovensku.sk/katlovce-kostol-sv-ducha (accessed on 19 October 2022).

33 This theme appeared frequently in respondents' statements. It is worth mentioning that the significant emigration from Upper Silesia to West Germany in the 1980s and 1990s contributed to providing tangible financial and material support for many families during the crisis caused by Poland's economic and political transformation.

34 From the perspective of Polish national identity, its constitutive determinant is the Polish language.

35 In news services broadcasting 24 h a day, the central theme was Russia's aggression against Ukraine.

36 https://www.ekai.pl/niemiecki-biskup-pod-wrazeniem-polskiej-pomocy-ukraincom/ (accessed on 9 November 2022).

37 https://synod.archidiecezjakatowicka.pl/synod/aktualnosci/210-synodalnosc-w-zyciu-i-misji-kosciola (accessed on 10 November 2022).

38 See Note 37.

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



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
