# Peer review of "The Roman Catholic Parish in the Face of the Ukrainian Refugee Crisis: A Case Study of St. Joseph Parish in Chorzów, Poland and Holy Spirit Parish in Kátlovce, Slovakia"

_religions, doi:10.3390/rel14081048_

Round 1
Reviewer 1 Report
Thanks for this original and timely research and data. This is a well-written research paper and its qualitative research is the strength of this paper. By reading this, readers will have a closer exposure to understanding how Russia's war against Ukraine has influenced the lives of ordinary people as well as faith communities.
However, while I understand that because the research is on such a new topic there is not much academic literature on the current issue of Russia's war against Ukraine, the author relies a lot on web resources. This is not to undervalue web resources but still speaks to the second issue, which is the lack of engagement with pastoral/spiritual care literature. The current draft reads more like a critical review of Catholic social teaching on the issues of migration.
The discussion section can be thickened. I suggest that the author would clarify more on how the reactions of the parishioners in the context of the guidelines of the social teaching of the Church exactly shape the way in which they engage in the work of pastoral care, especially in the discussion section. I think the discussion section can be improved by integrating quotations from the interviewers.
I appreciate reading this important work.
Author Response
We find the comments valuable however we cannot address the statement: "I am concerned that the writing is not clear in many places, which makes it difficult to understand", without pointing out specific aspects. In view of the observation that it is difficult to see that the social teaching of the church motivated the parishioners and at the same time that proof of this is needed, it must be said that the helpers in both countries were a community gathered around the church for many years, guided by certain values, and the very fact of their presence in the community of believers proves their influence. And this is how the special actors understand it, as will be supplemented in the text of the article. A detailed enumeration of the sequence of influence seems, with this explanation, less necessary. Does the reviewer's remark that: "Reference was made to the influence of parental upbringing, even though the largest group of participants was over 60 years old!" is a negative indication? If, even decades later, people are guided by their parents' upbringing, shouldn't this be particularly appreciated? The researchers inquired into what motivated the parishioners, among which was undoubtedly the teaching of the Catholic Church.
We would like to assure you that the next research process and especially the description of it will be improved, taking into account the valid comments of the reviewer, whom we sincerely thank. In our defence, it should be stated that the research was conducted in a most dynamic situation, in a difficult atmosphere, different from the usual design and conduct of research processes. It was also difficult to obtain statements in a matter that was often the personal, deep experience of a participant in a support group. We ask for your kind understanding of the work we undertook.
Reviewer 2 Report
This is a very timely study with an important research question. I'm afraid the writing is in many places not fully clear, impeding understanding. For example, the description of the research design is not clear, especially in the third paragraph. Other than the open-ended interviews, it isn't especially clear what the authors did.
As it stands, I am unable to see how the qualitative data indeed suggest that Catholic social teaching (at least as a body of principles addressed to social concerns) motivated the parishioners, which the authors suggest did in 60% of answers. The quotations used to support that conclusion suggest (as the authors note) religious motives rooted in love of neighbor, but only one or two suggest principles of Catholic social teaching such as solidarity. The authors need to make a clearer and better case that Catholic social teaching did motivate the parishioners, or at least what they mean by that.
That being said, this article is important in the way it chronicles the response of Slovakian and Polish Catholics to the Ukrainian refugee crisis, and its attempt to get at the reasons why people participated. Some of the answers as to why deserved more analysis, such as the reference to parental upbringing, or the overwhelming presence in the media, or the difference it made when people came to these towns. The parental upbringing influence was referenced even though the largest group of participants were over 60!
Were there any important differences between the two parishes?
Author Response
the reviewer rightly noted that reference could be made to the extensive 'pastoral/spiritual' literature. The present project is a more critical look at the reception of Catholic social teaching on migration. Indeed, the selection of literary evocations has been selective. The human drama caused by forced flight demanded this particular reference, or so it seemed to the authors of the text. The main objective was to reflect the information resource obtained in the interviews and the constraints of the publication volume indicated by the editors mean that we have concentrated on what is most relevant. We have also supplemented the text with the suggestions indicated, for which we sincerely thank you
Reviewer 3 Report
The research problem (oryginal and important) is well definied. The structure and the composition of the paper is correct. Methodology and the research goals are correct. The choice of literature is correct too, and fully related to the topic of the article. The article should be published as an original academic paper.
Author Response
We sincerely thank the reviewer who considered the article eligible for publication as it stands. It has not only analytical value, but also documentary value regarding a humanitarian crisis unprecedented in Europe on this scale for decades.